# Comorbid Hypertension Reduces the Risk of Ventricular Arrhythmia in Chronic Heart Failure Patients with Implantable Cardioverter-Defibrillators

**DOI:** 10.3390/jcm11102816

**Published:** 2022-05-17

**Authors:** Hao Huang, Yu Deng, Sijing Cheng, Nixiao Zhang, Minsi Cai, Hongxia Niu, Xuhua Chen, Min Gu, Xi Liu, Yu Yu, Wei Hua

**Affiliations:** 1State Key Laboratory of Cardiovascular Disease, Department of Cardiology, Fuwai Hospital, National Center for Cardiovascular Diseases, Chinese Academy of Medical Sciences and Peking Union Medical College, Beijing 100037, China; frankhuang1997@163.com (H.H.); ydeng18@163.com (Y.D.); chengsijing@fuwai.com (S.C.); zhangnx_doc@163.com (N.Z.); caiminsi2014@gmail.com (M.C.); drniu@126.com (H.N.); cheyne_xh@sina.com (X.C.); gumin1012@163.com (M.G.); fw16liuxi@foxmail.com (X.L.); yuyumd95@163.com (Y.Y.); 2Department of Cardiology, Cardiovascular Center, Beijing Friendship Hospital, Capital Medical University, Beijing 100050, China

**Keywords:** hypertension, systolic blood pressure, chronic heart failure, ventricular tachyarrhythmia

## Abstract

Aims: Low blood pressure (BP) has been shown to be associated with increased mortality in patients with chronic heart failure. This study was designed to evaluate the relationships between diagnosed hypertension and the risk of ventricular arrhythmia (VA) and all-cause death in chronic heart failure (CHF) patients with implantable cardioverter-defibrillators (ICD), including those with preserved left ventricular ejection fraction (HFpEF) and indication for ICD secondary prevention. We hypothesized that a stable hypertension status, along with an increasing BP level, is associated with a reduction in the risk of VA in this population, thereby limiting ICD efficacy. Methods: We retrospectively enrolled 964 CHF patients, with hypertension diagnosis and hospitalized BP measurements obtained before ICD implantation. The primary outcome measure was defined as the composite of SCD, appropriate ICD therapy, and sustained VT. The secondary endpoint was time to death or heart transplantation (HTx). We performed multivariable Cox proportional hazard regression and entropy balancing to calculate weights to control for baseline imbalances with or without hypertension. The Fine–Gray subdistribution hazard model was used to confirm the results. The effect of random BP measurements on the primary outcome was illustrated in the Cox model with inverse probability weighting. Results: The 964 patients had a mean (SD) age of 58.9 (13.1) years; 762 (79.0%) were men. During the interrogation follow-up [median 2.81 years (interquartile range: 1.32–5.27 years)], 380 patients (39.4%) reached the primary outcome. A total of 244 (45.2%) VA events in non-hypertension patients and 136 (32.1%) in hypertension patients were observed. A total of 202 (21.0%) patients died, and 31 (3.2%) patients underwent heart transplantation (incidence 5.89 per 100 person-years; 95% CI: 5.16–6.70 per 100 person-years) during a median survival follow-up of 4.5 (IQR 2.8–6.8) years. A lower cumulative incidence of VA events was observed in hypertension patients in the initial unadjusted Kaplan–Meier time-to-event analysis [hazard ratio (HR): 0.65, 95% confidence interval (CI): 0.53–0.80]. The protective effect was robust after entropy balancing (HR: 0.71, 95% CI: 0.56–0.89) and counting death as a competing risk (HR: 0.71, 95% CI: 0.51–1.00). Hypertension diagnosis did not associate with all-cause mortality in this population. Random systolic blood pressure was negatively associated with VA outcomes (*p* = 0.065). Conclusions: In hospitalized chronic heart failure patients with implantable cardioverter-defibrillators, the hypertension status and higher systolic blood pressure measurements are independently associated with a lower risk of combined endpoints of ventricular arrhythmia and sudden cardiac death but not with all-cause mortality. Randomized controlled trials are needed to confirm the protective effect of hypertension on ventricular arrhythmia in chronic heart failure patients.

## 1. Introduction

Chronic heart failure (CHF) is a common final stage of heart disease and represents a major cause of death and disability worldwide [1] and additionally, it is a health issue associated with increasing healthcare expenditures [2]. Hypertension, defined as blood pressure (BP) above 140/90 mmHg, is the leading risk factor for developing HF [3]. The clinical outcome is worse and mortality is increased in hypertensive patients with HF [4]. The hazard ratios (HRs) for developing HF in hypertensives compared with normotensives were twofold higher in men and threefold higher in women [5]. Therefore, BP management is a crucial step in the development of strategies that may prevent the progression of HF.

However, contrary to the well-established linear association between elevated BP and cardiovascular events in the general population [6], the association of hypertension and adverse outcomes in patients with established HF has been a topic of considerable discussion, as observational studies have found various types of J shaped, U shaped, and paradoxical negative linear relations [7,8,9,10,11]. These studies primarily linked single random BP measurements with clinical outcomes, which are affected by numerous cofounders, including underlying diseases and antihypertensive medication, leading to potential inaccuracy and uncertainty of these associations. Whether a pathological condition of stable hypertension exerts a protective role in the prognosis of CHF is still worth discussing.

Although death in patients with CHF is usually because of underlying cardiac disease, the cause-specific mechanisms could be split between sudden cardiac death (SCD) from arrhythmic events and non-sudden cardiac death (NSCD) because of pump failure [12]. To prevent the former cause of death in CHF patients, implantable cardioverter defibrillator (ICD) therapy is a widely accepted modality that can effectively monitor and terminate lethal ventricular arrhythmia [13]. Despite a recent meta-analysis reporting that prevalent hypertension and higher systolic and diastolic BP increase the risk of SCD among a diversified population [14], data on the association between hypertension and the risk of ventricular arrhythmias (VA) or ICD therapies in chronic heart failure patients, especially those with preserved left ventricular ejection fraction (LVEF), are still limited. It remains elusive whether and how exactly hypertension could predict VA in the HF population, or whether its protective effects only appear in low LVEF patients.

Therefore, the present study aimed to comprehensively evaluate the association of hypertension and blood pressure measurements on ventricular arrhythmic events in a real-world cohort of CHF patients with different functional statuses. We raise the following hypothesis that diagnosed hypertension and easily accessible BP measurements could help identify SCD high-risk populations who are eligible for a primary-prevention ICD.

## 2. Methods

At the time of regular device clinic follow-up visits, we enrolled sequential stable ambulatory patients with chronic heart failure implanted with a single/dual-chamber ICD between 1 January 2010 and 1 May 2020. Patients were included if they had at least one sign and one symptom of heart failure from a prespecified list of clinically defined signs and symptoms. For suspected heart failure patients, cardiac risk factors, the plasma concentration of NT-proBNP, chest X-ray, and echocardiography were comprehensively evaluated to confirm the diagnosis [15]. The exclusion criteria were (1) removal of the ICD within six months (*n* = 3), (2) without any interrogation follow-up beyond six months (*n* = 42), and (3) missing data on blood pressure and other significant variables (*n* = 13). Finally, 964 patients were included (Figure 1). The study was conducted in accordance with the Declaration of Helsinki and was approved by the Ethics Committee of Fuwai Hospital. Written informed consent was obtained from all patients before inclusion.

### 2.1. Data Collection and Device Programming

Demographic characteristics a physical examination and data on comorbidities, NYHA functional class, medication history, and laboratory testing results were collected from electronic medical records by trained clinicians at admission. Two-dimensional transthoracic echocardiography and laboratory tests were performed within 3 days before ICD implantation. BP was measured in a supine position and on both arms using an electronic sphygmomanometer by a physician on the day before ICD implantation during admission. Where there is a difference in BP between arms, the arm with the higher BP values was used for measurements. All measurements were to be read to the nearest 2 mmHg. The higher of the two readings was used for analysis.

ICD programming was standardized to eliminate variation in VA detection and therapy. After implantation, two-zone detection was programmed in the ICD: fast ventricular tachycardia (VT) (170–210 bpm) and ventricular fibrillation (>210 bpm). Supraventricular tachycardia discrimination algorithms were programmed for the VT zone. In all ICD patients, shock and anti-tachycardia pacing (ATP) during charging when possible were programmed in the ventricular fibrillation zone. The decision on programming ICD therapies in the fast VT zone was left to the discretion of the physician, which was primarily bursts of ATP followed by high-voltage shock(s) if ATP was unsuccessful.

### 2.2. Outcomes Measures

Clinical evaluation was performed before device implantation. The follow-up period started on day 1 after ICD implantation. Device interrogation including a review of the stored intracardiac electrograms was performed 3 months post-implantation and every 6–12 months thereafter. ATP and shocks were considered appropriate if the preceding rhythm was ventricular tachycardia or ventricular fibrillation (VT/VF). The primary outcome measure was defined as the composite of SCD, appropriate ICD therapy, and sustained VT. Sudden cardiac death was defined as unexpected death within 60 min of the onset of cardiac symptoms without prior cardiac deterioration, during sleep, or within ≤24 h of last being seen alive and clinically stable [16]. Board-certified electrophysiologists performed blinded adjudication of ICD therapy events. Inappropriate therapies were excluded from the outcomes. The secondary outcome measure was a combination of all-cause mortality or cardiac transplantation. The survival status was confirmed with medical death records or telephone calls until May 2021. The dates for the censoring of survival status and interrogation information are not necessarily the same.

### 2.3. Statistical Analysis

Continuous data are expressed as mean ± standard deviation or the median with the interquartile range (IQR) as appropriate; categorical data are presented as frequencies with percentages. Hypertension is defined as office SBP values ≥ 140 mmHg and/or DBP values ≥ 90 mmHg following repeated examination, and is classified as Grade 1 (SBP 140–159 mmHg and/or DBP 90–99 mmHg), Grade 2 (SBP 160–179 mmHg and/or DBP 100–109 mmHg), or Grade 3 (SBP ≥ 180 mmHg and/or DBP ≥ 110 mmHg) [4]. Baseline characteristics of patients with or without hypertension were compared using the unpaired *t*-test for normally distributed continuous variables, Kruskal–Wallis test for non-normally distributed continuous variables, and χ2 test for categorical variables.

The statistical analysis was performed in the following phases. First, we performed an analysis of the unadjusted cumulative incidence rates for both outcomes, illustrated by Kaplan–Meier time curves and a univariate Cox proportional hazard regression. Patients who were still free from a primary end-point event at the last interrogation visit, or who were still alive, were censored. The follow-up time was defined as the time between ICD device implantation (index date) and the outcome events or censoring. Patients without hypertension were the control group. An initial multivariable Cox regression model included demographic factors, clinical factors, medications, and laboratory tests. Cofounders were selected based on the Akaike information criterion (AIC) rule among univariable significant predictors with a *p*-value less than 0.1.

Second, to attenuate bias due to imbalances in baseline characteristics and to enhance our ability to draw inferences about the association, we used several matching and reweighting methods to assemble a cohort in which patients with or without hypertension would be expected to be balanced on all related baseline characteristics. Initially, propensity-score methods were used to reduce the effects of confounding. We estimated propensity scores for each of the 964 patients using a multivariable logistic regression model in which diagnosis of hypertension was used as the dependent variable and the 46 baseline variables were used as covariates. Associations between hypertension and ventricular arrhythmia were then estimated by multivariable Cox regression models with the use of three propensity-score methods. In the inverse-probability-weighted analysis, the predicted probabilities from the propensity-score model were used to calculate the stabilized inverse-probability-weighting weight. We conducted a secondary analysis that used propensity-score matching and another that included the propensity score as an additional covariate. In the propensity-score matching analysis, the nearest-neighbor method was applied to create a matched control sample.

To furtherly confirm our conclusions, we also performed an adjusted analysis using entropy balancing [17,18]. Entropy balancing is a reweighting method, which aims to produce an exact covariate balance of patients with and without hypertension. It is considered a generalization of propensity-score weighting and uses an optimization algorithm by assigning a scalar weight to each patient in the control group to balance means and variances between hypertension patients and the reweighted non-hypertension patients. In entropy balancing, no case is discarded. The estimated weights can be used as survey sampling weights in the subsequent analyses. Standardized differences were used for the balancing diagnostics instead of *p*-values [19]. A standardized difference >0.1 indicates a meaningful difference [20]. The entropy balancing weighting was used to calculate a weighted Kaplan–Meier curve for the non-hypertension patients and to perform a weighted multivariate Cox proportional hazard regression.

Then, formal sensitivity analyses were conducted to quantify the degree of hidden bias that could potentially explain any significant associations. First, a Fine–Gray subdistribution hazard model accounting for the competing risk of all-cause death was used to assess the association between hypertension and VA events in the same set of analyses. Next, the relationships of hypertension by different grades and corrected by random BP measurements before ICD implantation to the primary outcome were illustrated by a Kaplan–Meier time curve. Then, subgroup analyses were conducted to assess the homogeneity of the association between hypertension and primary outcome in clinically relevant subgroups of patients in the whole cohort. Finally, we explored the associations between pre-implantation random SBP or DBP measurements as continuous variables and the primary outcome. Potential nonlinearity was tested by using a likelihood ratio test comparing the model with only a linear term against the model with linear and cubic spline terms. Restricted cubic spline with best fit knots according to the AIC rule was used to flexibly model the potential nonlinear effects of SBP and DBP. Multiple imputation was used to handle missing data. The statistical analyses were performed using SPSS version 26.0 (for Windows, SPSS, Inc., Chicago, IL, USA) and R version 4.0.3 (R Foundation for Statistical Computing, Vienna, Austria, 2008). Two-sided *p*-value ≤ 0.05 was considered statistically significant if not otherwise specified.

## 3. Results

In total, the analysis included 964 patients with an ICD implantation, of whom 424 were diagnosed with hypertension. The baseline characteristics are displayed in Table 1. The population was predominantly male (79.0%) with a mean age of 58.9. The admission SBP and DBP levels were on average 10.4 and 4.3 mmHg higher in hypertension patients. Also, the hypertension group was more likely to suffer from diabetes mellitus, coronary arterial disease, stroke, hyperuricemia, hyperlipidemia, and renal dysfunction. They had higher left atrial diameters and higher left ventricular ejection fractions. More of these patients were taking RAAS inhibitors, CCB, statin, and antiplatelets, whereas fewer were taking MRA. The plasma levels of creatinine and erythrocyte sedimentation rates were higher in hypertension patients.

### 3.1. Primary Outcome

During the follow-up [median 2.81 years (interquartile range: 1.32–5.27 years)], 380 patients (39.4%) reached the primary outcome, 373 of whom had VT and ICD therapy and 7 (0.7%) of whom had SCD. A total of 244 (45.2%) VA events in non-hypertension patients and 136 (32.1%) in hypertension patients were observed. In the unadjusted Kaplan–Meier time-to-event curves, hypertension patients had a lower cumulative incidence of primary endpoints than non-hypertension patients (Figure 2A and Table 2) [hazard ratio (HR): 0.65, 95% confidence interval (CI): 0.53–0.80]. A multivariable model, adjusted for possible confounders in a univariate analysis, also showed a lower hazard ratio in the hypertension group (HR: 0.77; 95% CI: 0.61–0.96; Table 2).

In the multivariable analysis with inverse probability weighting according to the propensity score, there was still a significant association between hypertension diagnosis and the composite primary endpoint (HR: 0.73; 95% CI: 0.55–0.95; Table 2). Additional multivariable propensity-score analyses yielded similar results (Table 2). Furthermore, after the application of entropy balancing, the weighted average of the baseline characteristics of the hypertension patients was the same as that of the non-hypertension patients. Figure 3A shows the Kaplan–Meier curve for the primary outcome for the non-hypertension patients and the weighted Kaplan–Meier curves for the hypertension patients. A robust difference was found in the cumulative incidence of VA events. The hazard ratio for the primary composite endpoint, calculated using entropy-balanced multivariate Cox proportional hazard regression, was 0.71 (95% CI: 0.56–0.89). For both weighting methods, detailed information on the distribution of the baseline characteristics of hypertension patients according to the weight assigned to them is included in Appendix A. The between-group balance was assessed by estimating standardized mean differences for each of the 46 baseline characteristics and presented as a Love plot (Appendix A).

### 3.2. Secondary Outcome and Sensitivity Analysis

During a median survival follow-up of 4.5 (IQR 2.8–6.8) years, 202 (21.0%) patients died, and 31 (3.2%) patients underwent heart transplantation (incidence 5.89 per 100 person-years; 95% CI: 5.16–6.70 per 100 person-years). The same set of analyses of the primary outcome was applied. No significant difference in all-cause mortality could be found in chronic heart failure patients with or without hypertension (Table 2 and Figure 2B and Figure 3B).

In the sensitivity analyses, the results were virtually identical. A competing risk model taking premature death into account demonstrated a virtually identical result (Appendix A). When refining the hazard ratios across different hypertension grades, for controlled or uncontrolled hypertension (random SBP ≥ 140 mmHg or DBP ≥ 90 mmHg) the protective effect of hypertension was consistent (Appendix A). The findings from our subgroup analyses demonstrate that the association between hypertension and VA events was homogenous across various clinically relevant subgroups of patients, except for those with ischemic etiology (Figure 4). The findings from our restricted cubic spline analysis demonstrated that there was no evidence of a nonlinear association between the SBP and VA events in both the propensity score-adjusted and inverse probability weighted cohorts (*p* > 0.10, Figure 5), though a possible negative relationship was detected in the weighted cohort (*p* = 0.065). For DBP, a significant nonlinearity was found in the weighted cohort, with the highest risk of primary endpoint reached at around 70 mmHg (*p* for nonlinearity = 0.004, Appendix A).

## 4. Discussion

In this study, we mainly evaluated the prognostic value of hypertension diagnosis for a better selection of ICD candidates in CHF patients. The findings from the current study demonstrate that among hospitalized CHF patients eligible for an ICD, the comorbidities of hypertension and higher SBP levels were associated with a significantly lower risk of ventricular arrhythmic events, though without increasing the mortality risk. These findings, taken together with multiple sensitivity analyses, provide strong evidence of a consistent association between hypertension and better ventricular arrhythmia outcomes in patients with CHF.

Apart from all-cause or cardiac mortality, ventricular tachyarrhythmias and SCD are important outcomes that have been underreported in CHF patients. Although evaluating the risk of cardiac mortality provides prognostic information, cost-effective decision making with regard to ICD therapy requires approaches to discriminating patients with a high risk of SCA from those more likely to succumb to heart failure and pump dysfunction. LVEF was used as an important index for SCD risk stratification, given that patients with HFrEF are at an increased risk of VA and SCD irrespective of HFrEF etiology [21], but it still lacks sensitivity as a prognostic marker. On the other hand, studies exploring noninvasive risk factors for SCD in patients with HFpEF do not identify consistent factors except for ischemic heart disease [22]. Consequently, there is no accepted noninvasive test to identify high-risk patients with HFpEF.

Hypertension and elevated SBP have been described as surrogate risk markers for VA events in general population studies [14,23]. In contrast to patients with normal cardiac function, previous studies have shown an inverse correlation between SBP and the risk of adverse events, mostly death, in CHF patients with LVEF reduced or preserved [10,24,25]. The protective effect of CHF could be extended to ventricular arrhythmia, which was previously detected by post hoc analysis of MADIT-RIT and MADIT-II trials in which elevated SBP are at a lower risk of VA and SCD [26,27]. However, much confusion remains. First, many studies categorized patients into 2–4 groups based on a single random BP measurement, which may not reflect a certain hypertension status [9,10,11,26,27]. Second, we noticed significant differences in the baseline characteristics between randomly SBP-grouped cohorts since this is the inherent vice of a non-randomized-control design. Third, previous studies only targeted patients with a lower LVEF. Last but not least, when applying the conventional Cox proportional risk model, one may wonder if the increase in the competing risk of pump failure death in patients with hypertension may be the actual culprit of an observed reduced incidence of VA events. Given the existing knowledge gap, our study gave an elaborate verification in agreement with the previous ones. As an innovative element, we attempted to mimic an RCT by applying entropy balancing to render the diagnosis of hypertension independent of all the measured covariates including random BP levels. Other possible determinants significant in the univariate analysis were also included in multivariable models. A robust conclusion could be drawn when repeating the weighting analysis accounting for death as a competing risk.

Contrary to conventional perceptions, we raised some hypotheses about the reverse epidemiology of hypertension and sudden cardiac arrest in CHF patients, similar to the reverse association of blood pressure and all-cause death widely reported in established heart failure. On the one hand, hypertension is the most common cause of hypertensive heart disease, which comprises left ventricular hypertrophy (LVH), left atrial enlargement, diastolic dysfunction, functional mitral regurgitation, and neurohormonal changes, in which LVH has a well-established relationship with SCD [28,29]. Large clinical trials and meta-analyses have proven that hypertension is a major risk factor for SCD as well as cardiovascular disease [14,30,31] and that effective BP control and regression of LVH during antihypertensive therapy was associated with a 30% lower risk of SCD, independently of blood pressure lowering and other known predictors of SCD [32]. However, on the other hand, these conclusions are all confirmed in the general population, in whom gradually developing cardiac structural abnormalities, including hypertrophy and fibrosis, seem to be the direct cause of SCD, instead of hypertension per se. Ischemic etiology is blamed for a large proportion of SCD. Although the Framingham Study found that elevated blood pressure is a powerful independent predisposing factor for all clinical manifestations of coronary heart disease, including sudden death, the proportion of sudden coronary heart disease deaths is no greater in hypertensive than in normotensive persons once overt coronary heart disease becomes manifest in symptomatic individuals [33]. More recent studies have also found that ventricular tachycardia or ventricular fibrillation (VT/VF) after MI was associated with lower blood pressure [34,35]. In hypertrophic cardiomyopathy, an abnormal blood pressure response (failure to increase SBP or decreased SBP) during exercise testing has been postulated to be a risk factor for SCD [36]. All the above evidence demonstrates the potential risks of low blood pressure in both ischemic cardiomyopathy (ICM) and non-ischemic cardiomyopathy (NICM). Our study demonstrated that the diagnosis of hypertension referred by untreated blood pressure ruled out the antihypertensive effect and may reflect an earlier cardiac function before the final development of CHF. In that sense, hypertension is not necessarily arrhythmogenic per se but indicates better myocardial reserve and coordinated ventricular depolarization, which translates into fewer VA. When ventricles grow larger and failed, elevated blood pressure may have a lesser effect on the ventricle in terms of shear stress and overload. In contrast, the relatively low SBP, despite increased sympathetic activity in CHF patients, plays a significant role in the electrical dysfunction of the heart. This imbalance could lead to the generation of ventricular arrhythmia [37].

In addition, a higher proportion of pre-weighted patients with hypertension were receiving ACEI/ARB and CCB (almost dihydropyridine CCB), which may have contributed to better-controlled reflex neurohormonal activation, and therefore better outcomes. Although entropy balancing achieved substantial between–group balance in all measured confounders, imbalances in their severity may remain and persist during follow-up. Furthermore, imbalances in unmeasured confounders may also, in part, explain the observed better outcomes in the hypertension group.

In the subgroup analysis, it is interesting to find that hypertension reduced the VA risk more evidently in ICM patients rather than in NICM (*p* for interaction = 0.004), which is in accordance with the previous study [27]. More importantly, we found a consistent protective effect of hypertension throughout different LVEF groups and ICD prevention indications. This can help to extrapolate the conclusion to HFpEF patients and the ICD secondary prevention population (Figure 4). To the best of our knowledge, we have confirmed for the first time that hypertension could protect against ventricular arrhythmic events in HFpEF patients. Although the implantation of an ICD appears to be mandatory among patients with suspected sustained VT/VF history, a diagnosed hypertension status may predict less ICD therapy or interrogation frequency, thus improving quality of life once heart failure has been established.

The hazard ratios and 95% confidence intervals for the primary VA outcome by systolic blood pressure level in 964 patients with chronic heart failure according to restricted cubic spline regression models using 3 knots were calculated. Solid black lines indicate hazard ratios and shaded areas indicate 95% CI. Plots on the left panel (A) are adjusted for propensity scores, and those on the right panel (B) are inverse probability-weighted on 46 baseline characteristics.

In most studies, systolic BP (SBP) but not diastolic (DBP) was shown to directly correlate with the risk of cardiovascular disease. This phenomenon was also observed in our multiple sensitivity analysis, in which SBP showed a negative linear association with VA outcomes (*p* for linear = 0.065). What surprised us is that DBP could demonstrate a possible converse U-shape association with VA (*p* for nonlinear = 0.004). We suspected that potential overfitting may exist when most single DBP measurements cluster around the median. Also, we can draw robust conclusions when dividing those diagnosed with hypertension into daily controlled or uncontrolled (random SBP ≥ 140 mmHg or DBP ≥ 90 mmHg) groups (Appendix A). All the above evidence shows that the diagnosis of hypertension serves as a better marker for predicting VA events rather than single BP measurements, and also that SBP weighs more than DBP.

Overall, our analysis comprises a relatively long timeframe with a large number of patients in a real-world setting and reflects contemporary therapy. It is distinguished by the use of inverse probability weighting and entropy balancing to assemble a balanced cohort, the use of subgroup analyses to demonstrate homogeneity, and the use of multiple sensitivity analyses by diversified statistical models and grouping adjustments to assess bias. Thus, our findings may suggest that comorbid hypertension, along with elevated SBP, should be taken into account for improving risk stratification, as they are at a significantly lower risk for VA. Perhaps in those patients, the benefits of primary defibrillator implantation are more limited.

However, several limitations are worth noting. First, to account for SCA events in subjects with an ICD, we included ICD discharges for ventricular fibrillation or fast ventricular tachycardia (>170 bpm). A more restrictive threshold would decrease the frequency of arrhythmic events (potential SCA equivalents) and could alter the results of this analysis. In addition, the lack of longitudinal measurements of blood pressure could underestimate its prognostic role in predicting outcomes. In other words, longitudinal measurements of blood pressure and electrophysiological properties are needed to investigate their relationship, both before the onset and during the worsening of heart failure.

## 5. Conclusions

In hospitalized chronic heart failure patients with implantable cardioverter-defibrillators, hypertension status and higher systolic blood pressure measurements are independently associated with a lower risk of combined endpoints of ventricular arrhythmia and sudden cardiac death, but not of all-cause mortality. This may be useful for identifying lower-risk patients, in whom the benefits of primary defibrillator implantation are more limited.

## Figures and Tables

**Figure 1 jcm-11-02816-f001:**
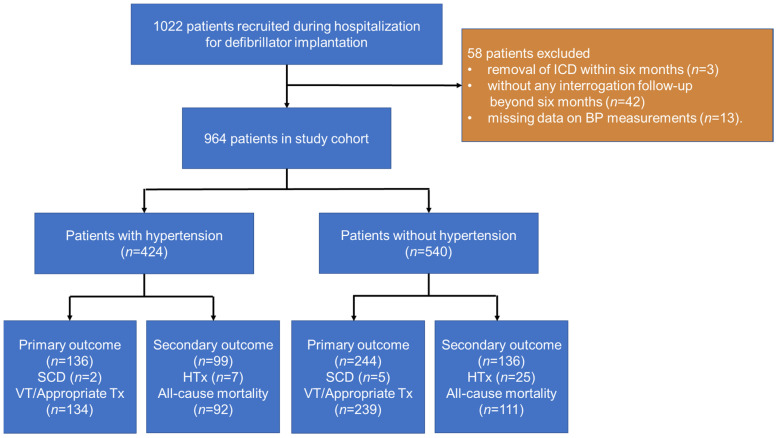
Flow chart depicts patients included in the study and outcomes. Appropriate Tx, appropriate ICD therapy; BP, blood pressure; HTx, heart transplantation; ICD, implantable cardioverter-defibrillator.

**Figure 2 jcm-11-02816-f002:**
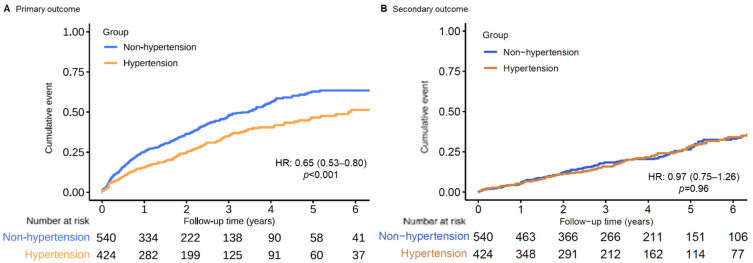
Unadjusted Kaplan–Meier time-to-event curves for the cumulative incidence of (**A**) primary and (**B**) secondary outcome events for patients with and without hypertension.

**Figure 3 jcm-11-02816-f003:**
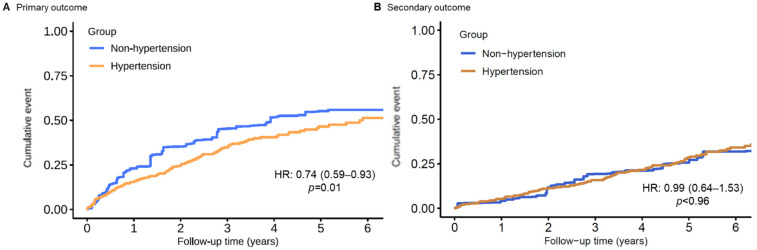
Entropy-balancing weighted Kaplan–Meier time-to-event curves for the cumulative incidence of (**A**) primary and (**B**) secondary outcome events for patients with and without hypertension.

**Figure 4 jcm-11-02816-f004:**
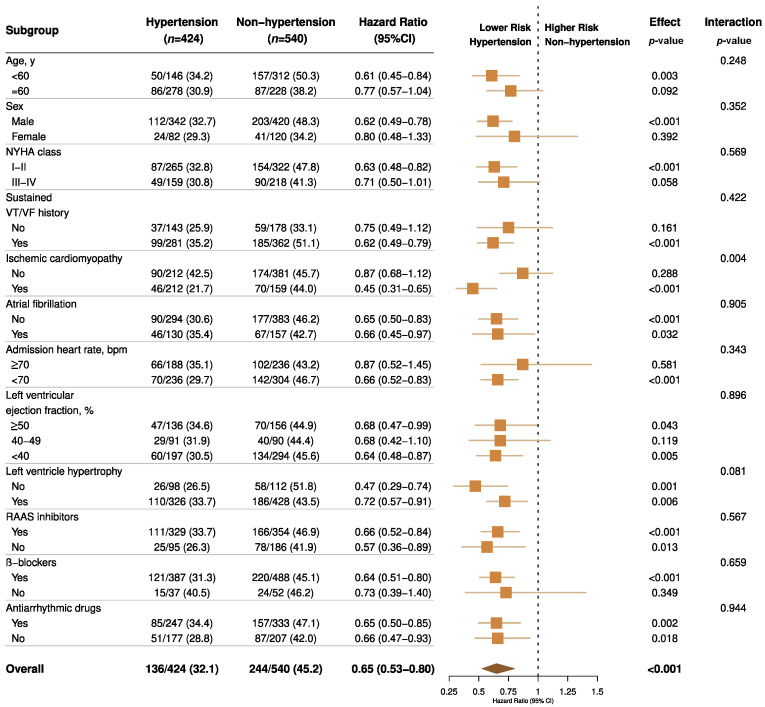
Forest plots for subgroup analyses of primary outcome by hypertension status. Forest plots displaying hazard ratios and 95% confidence intervals for primary VA outcome in subgroups of patients with heart failure by hypertension status. NYHA, New York Heart Association; VT/VF indicates ventricular tachycardia/fibrillation; RAAS, renin-angiotensin-aldosterone system.

**Figure 5 jcm-11-02816-f005:**
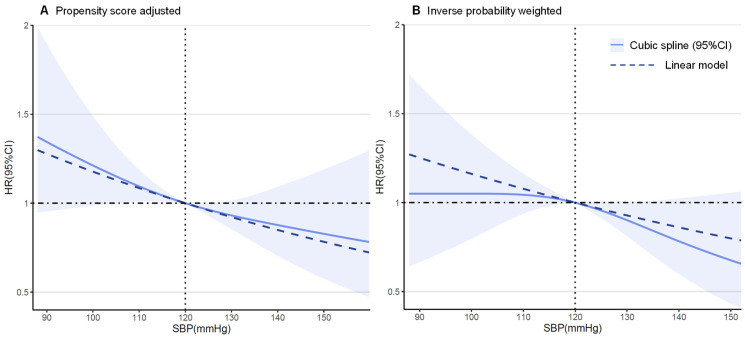
Restricted cubic spline plots for primary outcome by systolic blood pressure (SBP).

**Table 1 jcm-11-02816-t001:** Baseline Characteristics by Hypertension diagnosis in Patients with Chronic Heart Failure.

Characteristic	No Hypertension (*n* = 540)	Hypertension (*n* = 424)	*p*-Value
Age	55.6 ± 13.8	63.1 ± 10.7	**<0.001**
Male	420 (77.8)	342 (80.7)	0.312
Body mass index (kg/m^2^)	24.2 ± 3.3	25.6 ± 3.7	**<0.001**
Heart rate (bpm)	69.0 ± 13.9	68.5 ± 13.9	0.567
Systolic blood pressure	115.1 ± 14.5	125.5 ± 16.0	**<0.001**
Diastolic blood pressure	71.3 ± 9.3	75.6 ± 10.8	**<0.001**
NYHA class			0.401
I/II	322 (59.6)	265 (62.6)	
III/IV	218 (40.4)	159 (37.5)	
Smoking	239 (44.3)	203 (47.9)	0.292
Alcohol use	185 (34.3)	165 (38.9)	0.154
SCD family history	26 (4.8)	21 (5.0)	1.000
ICD Primary prevention	178 (33.0)	143 (33.7)	0.857
Dual-chamber	190 (35.2)	163 (38.4)	0.330
Syncope	251 (46.5)	170 (40.1)	0.055
Ablation history	53 (9.8)	30 (7.1)	0.165
**Medical history**			
Diabetes mellitus	72 (13.3)	121 (28.5)	**<0.001**
Atrial fibrillation	157 (29.1)	130 (30.7)	0.643
Atrioventricular block	68 (12.6)	53 (12.5)	1.000
Coronary arterial disease	196 (36.3)	262 (61.8)	**<0.001**
Stroke	24 (4.4)	38 (9.0)	**0.007**
Pulmonary hypertension	46 (8.5)	36 (8.5)	1.000
Hyperuricemia	45 (8.3)	55 (13.0)	**0.025**
Hyperlipidemia	190 (35.2)	289 (68.2)	**<0.001**
Frequent PVCs	233 (43.1)	192 (45.3)	0.514
eGFR < 60 mL/min/1.73 m^2^	90 (16.7)	139 (32.8)	<0.001
**Echocardiographic Parameters**			
Left atrial diameter	43.1 ± 8.4	44.2 ± 7.3	**0.029**
Left ventricular mass index	151.0 ± 54.4	150.2 ± 50.5	0.811
Right ventricular diameter	23.5 ± 5.9	23.2 ± 4.5	0.383
Left ventricular ejection fraction	40.8 ± 14.1	42.8 ± 13.3	**0.026**
HFrEF	294 (54.4)	197 (46.5)	
HFmrEF	90 (16.7)	91 (21.5)	
HFpEF	156 (28.9)	136 (32.1)	
**Medications**			
Antiarrhythmic drugs	333 (61.7)	247 (58.3)	0.314
ACEI/ARB	327 (60.6)	316 (74.5)	**<0.001**
ARNI	27 (5.0)	13 (3.1)	0.146
β-blocker	488 (90.4)	387 (91.3)	0.712
Calcium channel blockers	20 (3.7)	76 (17.9)	**<0.001**
Loop diuretics	397(73.5)	299 (70.5)	0.337
Mineralcorticoid receptor antagonist	378 (70.0)	255 (60.1)	**0.002**
Digoxin	129 (23.9)	107 (25.2)	0.684
Statin	231 (42.8)	272 (64.2)	**<0.001**
Anticoagulants	103 (19.1)	93 (21.9)	0.295
Antiplatelets	152 (28.1)	186 (43.9)	**<0.001**
**Laboratory Parameters**			
NT-proBNP (ng/mL)	882.8 (390.8, 1763.8)	968.9 (392.4, 2167.6)	0.346
Hemoglobin (g/L)	143 (132, 153)	142 (129, 154)	0.478
LDH (U/L)	189 (161, 225)	189 (161, 228)	0.821
ESR (mm/h)	6 (3, 13)	7.5 (3, 14)	**0.033**
TC (mmol/L)	3.99 (3.36, 4.79)	3.81 (3.20, 4.72)	0.052
LDL (mmol/L)	2.34 (1.79, 3.09)	2.25 (1.75, 2.94)	0.307
HDL (mmol/L)	1.06 (0.88, 1.24)	0.97 (0.83, 1.18)	**0.016**

ACEI/ARB, angiotensin-converting enzyme inhibitor/angiotensin receptor blocker; ARNI, Angiotensin receptor neprilysin inhibitor; ESR, erythrocyte sedimentation rate; HDL, high-density lipoprotein cholesterol; ICD, implantable cardioverter-defibrillator; LDH, lactic dehydrogenase; LDL, low-density lipoprotein; MRA, mineralocorticoid receptor antagonist; NT-proBNP, N-terminal pro-B-type natriuretic peptide; NYHA, New York Heart Association; PVCs, premature ventricular complexes; SCD, sudden cardiac death; TC, total cholesterol. Bold as *p*-value less than 0.5.

**Table 2 jcm-11-02816-t002:** Associations between Hypertension diagnosis and the primary composite endpoint of VA and all-cause mortality in the crude analysis, multivariable analysis, propensity-score analysis, and entropy-balanced analysis.

Analysis	Primary Composite Outcome HR (95% CI)	*p*-Value	All-Cause Mortality HR (95% CI)	*p*-Value
Crude analyses	0.65 (0.53–0.80)	**<0.001**	0.97 (0.75–1.26)	0.817
Multivariable analyses *	0.77 (0.61–0.96)	**0.023**	0.89 (0.67–1.17)	0.391
Propensity-score analyses				
With inverse probability weighting ^†^	0.73 (0.55–0.95)	**0.022**	0.87 (0.63–1.21)	0.417
With matching ^‡^	0.71 (0.52–0.96)	**0.026**	0.95 (0.67–1.34)	0.759
Adjusted for propensity score ^§^	0.75 (0.58–0.97)	**0.028**	0.94 (0.69–1.29)	0.694
Entropy-balanced weighting analyses ^※^	0.69 (0.49–0.98)	**0.036**	0.96 (0.61–1.51)	0.874

* Shown is the hazard ratio from the multivariable Cox proportional hazards model, with additional adjustment for all demographic characteristics, comorbidities, echocardiographic parameters, medications, and laboratory parameters. Hypertension, together with age, sex, ICD prevention indication, coronary atrial disease, pulmonary hypertension, right ventricular diameter, and calcium channel blockers remained in the final model. The analysis included all 964 patients. ^†^ Shown is the primary analysis with a hazard ratio from the multivariate Cox proportional hazards model adjusted with the same covariates as inverse probability weighting according to the propensity score. The analysis included all the patients. ^‡^ Shown is the hazard ratio from a multivariable Cox proportional hazards model with the same covariates matching according to the propensity score. The analysis included 482 patients (241 with hypertension and 241 without). ^§^ Shown is the hazard ratio from a multivariable Cox proportional hazards model with additional adjustment for the propensity score. The analysis included all the patients. ^※^ Shown is the hazard ratio from a multivariable Cox proportional hazards model using weights from entropy balancing. The analysis included all the patients. Bold as *p*-value less than 0.5.

## Data Availability

Not available.

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
