# Peer review of "Comorbid Hypertension Reduces the Risk of Ventricular Arrhythmia in Chronic Heart Failure Patients with Implantable Cardioverter-Defibrillators"

_jcm, 2022, doi:10.3390/jcm11102816_

Round 1
Reviewer 1 Report
The paper is a well conducted study that confirms a paradox. Despite the hazard ratios for developing HF and cardiac death in hypertensives compared with normotensives were 2-fold higher in men and 3-fold higher in women, the association of hypertension and adverse outcomes in patients with established HF has been previously demonstrated, and has been the topic of important discussion.
As pointed out , the cause-specific mechanism of Cardiac death in patients with CHF could be split between sudden cardiac death (SCD) from arrhythmic events and non-sudden cardiac death (NSCD) because of pump failure. The study was mainly devoted to study the link between hypertension and SD in this large group of patients with CHF and ICD implantation. At a limited follow up period (2.8 years), 39.4% of the entire group suffered from a cardiac ventricular arrhythmia, indicating an appropriate selection of patients truly at risk.
The study protocol and the subsequent follow up were well conducted and the statistic analysis remarkable.
It is of interest that the hypertension group was more likely to suffer from diabetes mellitus, coronary arterial disease, stroke, hyperuricemia, hyperlipidemia and renal dysfunction. They had higher left atrial diameters and higher left ventricular ejection fraction. The plasma level of creatinine and erythrocyte sedimentation rate were also higher in hypertension patients. Despite this worst medical status the hypertensive group suffered from less arrhythmic events (32% versus 45%), but had a similar rate of all-cause mortality.
The study demonstrated that hypertension is not necessarily arrhythmogenic per se, but indicates better myocardial reserve and coordinate ventricular depolarization, which translates into less VA. The treatment with ACE / CCB (almost dihydropyridine CCB), may have contributed to better-controlled reflex neurohormonal activation, and therefore better outcomes.
In conclusion the authors have presented relevant data on the protective effect of hypertension in a relevant number of patients with HFpEF. It will be of interest in the future to know if it is possible to ascertain the subgroup of patients in whom an ICD implantation is not necessary.
Author Response
Dear reviewer,
Thank you for your careful review! Please check out the revised manuscript below.
Best Regards,
Wei Hua
Reviewer 2 Report
Authors:
Dear Sir/Madam,
I had the opportunity to act as a reviewer on the recent submission by Huang et al. to the Journal of Clinical Medicine.
The authors present interesting original research regarding the risk of ventricular arrhythmias in chronic heart failure patients with implantable cardioverter defibrillators and comorbid hypertension. They have retrospectively included a total of 964 chronic heart failure patients and found that higher systolic blood pressure measurements are independently associated with a lower risk of combined endpoints of ventricular arrhythmia and sudden cardiac death.
The manuscript is well written and the results are interesting and of high clinical interest.
However, some issues need to be addressed:
- I recommend using a single guideline for hypertension throughout the text: i.e., the guideline from the ESC. Regarding this matter, I recommend erasing the word “commonly” on line 52, as this is according to the guideline the actual definition.
- Why were the patients with CRT implantations not included? Did this represent an exclusion criterion? If so, why? Please explain (see line 88).
- Please add in the baseline characteristics (Table 1), as in Fig. 4, the number of patients with HFpEF, HFmrEF and HFrEF (using standard definitions according to the ESC guideline). Furthermore, please add the left atrial volume index (LAVI) under the echocardiography parameters.
- The authors mention in the Discussion section that “a higher target of SBP level may help to reduce ICD therapy to improve quality of life once heart failure has been established.” Does this mean withholding HF substances such as ARNIs or ACE inhibitors just to keep the blood pressure high? It has been already shown that the ARNIs reduce the VA burden. Please comment on this.
- I strongly recommend introducing a sentence in the conclusion stating that to definitely answer the question RCT data are needed. Would the authors based on their study refuse to implant an ICD in a patient with hypertension under HF therapy and ejection fraction 30% (primary prophylaxis)?
- I recommend removing the word “of” on line 384.
- Please add the legend to the figure S1 (EB, IPTW).
Best regards,
Author Response
Dear Reviewer,
Thank you for your kind and informative comments! We did some revisions according to your advice, and here are some responses to your questions.
- I recommend using a single guideline for hypertension throughout the text: i.e., the guideline from the ESC. Regarding this matter, I recommend erasing the word “commonly” on line 52, as this is according to the guideline the actual definition.
Checked. Thank you for your advice!
- Why were the patients with CRT implantations not included? Did this represent an exclusion criterion? If so, why? Please explain (see line 88).
Patients with CRT implantations share different characteristics than those with ICD. They are more likely to suffer from left bundle branch block and prolonged QRS, which implies not only a more sensitive response to moderate elevation of arterial pressure, reflected by marked reductions in left ventricular ejection fraction (LVEF) and global longitudinal strain (GLS)1, but also more prevalence of sudden cardiac death and ventricular arrhythmia2,3. More importantly, CRT is only applied in heart failure with reduced ejection fraction (HFrEF), which means other types of HF including HFmrEF and HFpEF are underrepresented4. Contrary to ICD, the different response rate of cardiac resynchronization therapy and its direct impact on blood pressure elevation would confound the effect of diagnosed hypertension on VA, as a meta-analysis has reported super-responders have a lower absolute risk of VA and all-cause mortality5. Previous study has shown the relationship between baseline BP level and outcomes in patients with CRT, in which lower baseline SBP value is associated with incremental clinical benefits6. This study was initially designed for detecting risk factors of ventricular arrhythmia events in chronic heart failure patients. We believe that including ICD population was eligible and representative to draw this conclusion.
- Please add in the baseline characteristics (Table 1), as in Fig. 4, the number of patients with HFpEF, HFmrEF and HFrEF (using standard definitions according to the ESC guideline). Furthermore, please add the left atrial volume index (LAVI) under the echocardiography parameters.
Baseline characteristics of heart failure categories have been added. We regret that due to retrospective design, left atrial volume index was mostly unavailable in our database. However, we believe left atrial diameter was a matchable variable that would not affect the robustness of the conclusion.
- The authors mention in the Discussion section that “a higher target of SBP level may help to reduce ICD therapy to improve quality of life once heart failure has been established.” Does this mean withholding HF substances such as ARNIs or ACE inhibitors just to keep the blood pressure high? It has been already shown that the ARNIs reduce the VA burden. Please comment on this.
We might cause a misunderstanding here in the discussion section. In most post-hoc analyses on device implantation for heart failure, the baseline BP level was actually determined as a consequence of guideline-directed management and therapy (GDMT). In this study, we argue that the prognostic value of a single random SBP under GDMT is not as competent as diagnosed hypertension, in which case we would not recommend down-titration based on that single value.
On the other hand, although the evidence of current optimal medication treatment mainly based on outcomes such as all-cause mortality and heart failure hospitalization was robust, their effect on arrhythmic events was underreported. Growing evidence has shown ARNIs reduce the VA burden in HFrEF patients but the mechanism is still not utterly clear. It is suggested that its anti-arrhythmic effect is largely attributed to left ventricular reverse remodeling rather than lowering blood pressure7, and perhaps the former one takes advantage. Given the number of patients taking ARNI in this retrospective cohort was pretty low (4.1%), Further studies are needed to explore the possible counterpart mechanism on VA in this new class of drugs.
- I strongly recommend introducing a sentence in the conclusion stating that to definitely answer the question RCT data are needed. Would the authors based on their study refuse to implant an ICD in a patient with hypertension under HF therapy and ejection fraction 30% (primary prophylaxis)?
Thank you for your advice! We would prefer to selectively reduce ICD implantation or interrogation frequency in a proportion of patients with hypertension for primary prophylaxis. However, given that still a quarter of this population experienced VA events during long-term follow-up in the study, other residual risk factors must be taken into consideration.
- I recommend removing the word “of” on line 384.
Checked.
- Please add the legend to the figure S1 (EB, IPTW).
Checked.
Reference
- Aalen J, Storsten P, Remme EW, et al. Afterload Hypersensitivity in Patients With Left Bundle Branch Block. JACC Cardiovascular imaging 2019; 12(6): 967-77.
- Rabkin SW, Mathewson FA, Tate RB. Natural history of left bundle-branch block. British heart journal 1980; 43(2): 164-9.
- Tan NY, Witt CM, Oh JK, Cha YM. Left Bundle Branch Block: Current and Future Perspectives. Circulation Arrhythmia and electrophysiology 2020; 13(4): e008239.
- McDonagh TA, Metra M, Adamo M, et al. 2021 ESC Guidelines for the diagnosis and treatment of acute and chronic heart failure. European heart journal 2021; 42(36): 3599-726.
- Yuyun MF, Erqou SA, Peralta AO, et al. Risk of ventricular arrhythmia in cardiac resynchronization therapy responders and super-responders: a systematic review and meta-analysis. Europace : European pacing, arrhythmias, and cardiac electrophysiology : journal of the working groups on cardiac pacing, arrhythmias, and cardiac cellular electrophysiology of the European Society of Cardiology 2021; 23(8): 1262-74.
- Biton Y, Moss AJ, Kutyifa V, et al. Inverse Relationship of Blood Pressure to Long-Term Outcomes and Benefit of Cardiac Resynchronization Therapy in Patients With Mild Heart Failure: A Multicenter Automatic Defibrillator Implantation Trial With Cardiac Resynchronization Therapy Long-Term Follow-Up Substudy. Circulation Heart failure 2015; 8(5): 921-6.
- Vecchi AL, Abete R, Marazzato J, et al. Ventricular arrhythmias and ARNI: is it time to reappraise their management in the light of new evidence? Heart failure reviews 2022; 27(1): 103-10.